# Graphene Quantum Dots in Bacterial Cellulose Hydrogels for Visible Light-Activated Antibiofilm and Angiogenesis in Infection Management

**DOI:** 10.3390/ijms26031053

**Published:** 2025-01-26

**Authors:** Danica Z. Zmejkoski, Nemanja M. Zdravković, Dijana D. Mitić, Zoran M. Marković, Milica D. Budimir Filimonović, Dušan D. Milivojević, Biljana M. Todorović Marković

**Affiliations:** 1Vinča Institute of Nuclear Sciences—National Institute of the Republic of Serbia, University of Belgrade, P.O. Box 522, 11001 Belgrade, Serbia; zoranmarkovic@vin.bg.ac.rs (Z.M.M.); mickbudimir@gmail.com (M.D.B.F.); dusanm@vinca.rs (D.D.M.); 2Scientific Institute of Veterinary Medicine of Serbia, Janisa Janulisa 14, 11107 Belgrade, Serbia; nemanja.zdravkovich@gmail.com; 3Faculty of Dental Medicine, University of Belgrade, Dr. Subotića 8, 11000 Belgrade, Serbia; dijana.trisic@stomf.bg.ac.rs

**Keywords:** graphene quantum dots, bacterial cellulose, photodynamic therapy, antibiofilm, human gingival fibroblasts, angiogenesis, singlet oxygen

## Abstract

A novel bacterial cellulose (BC)-based composite hydrogel with graphene quantum dots (BC-GQDs) was developed for photodynamic therapy using blue and green light (BC-GQD_blue and BC-GQD_green) to target pathogenic bacterial biofilms. This approach aims to address complications in treating nosocomial infections and combating multi-drug-resistant organisms. Short-term illumination (30 min) of both BC-GQD samples led to singlet oxygen production and a reduction in pathogenic biofilms. Significant antibiofilm activity (>50% reduction) was achieved against *Staphylococcus aureus* and *Escherichia coli* with BC-GQD_green, and against *Pseudomonas aeruginosa* with BC-GQD_blue. Atomic force microscopy images revealed a substantial decrease in biofilm mass, accompanied by changes in surface roughness and area, further confirming the antibiofilm efficacy of BC-GQDs under blue and green light, without any observed chemical alterations. Additionally, the biocompatibility of BC-GQDs was demonstrated with human gingival fibroblasts (HGFs). For the first time, in vitro studies explored the visible light-induced potential of BC-GQD composites to promote wound healing processes, showing increased migratory potential and the upregulation of *eNOS* and *MMP9* gene expressions in HGFs. Chemical characterization revealed a 70 nm upshift in the photoluminescence emission spectra compared to the excitation wavelength. These novel photoactive BC-GQD hydrogel composites show great promise as effective agents for wound healing regeneration and infection management.

## 1. Introduction

Nosocomial infections develop within a healthcare facility and significantly contribute to patient morbidity, mortality, and financial strain on healthcare systems. These infections include urinary tract infections (especially catheter-associated) (12.9%), respiratory pneumonia (ventilator-associated) (21.8%), skin and gastrointestinal infections (17.1%), and surgical site wound infections (21.8%) [1]. A key characteristic of nosocomial infections is their development at least 48 h after hospital admission. Earlier studies [2,3] report that 3.2% of all patients in the USA and 6.5% in the EU acquire hospital-associated infections. The occurrence is even higher worldwide. Klevens et al. [4] reported the highest rates among adults and children in and outside intensive care units, high-risk neonates, and well-baby nurseries.

The primary transmission route for nosocomial infections is through direct or indirect contact. Complications in treating these infections arise from multi-drug-resistant organisms, which represent around 20% of all reported pathogens [5]. Rated as one of the top 10 global public health threats facing humanity, antimicrobial resistance occurs when microorganisms develop a mechanism to resist the medicines that are relied upon for treatment, worsening some conditions until they are difficult or impossible to cure [6].

The widespread emergence of extensive drug-resistant bacteria, including multi-drug-resistant strains, presents a daring issue in developing new antibiotic alternatives. When host immune system breaks down, bacteria from the natural flora may cause infections. The formation of bacterial biofilms further complicates medical treatment, increasing bacterial tolerance to antimicrobials and facilitating gene transfer. All this results in higher mortality and morbidity rates in patients with wounds [7,8].

The most prevailing Gram-positive bacteria include *Staphylococcus aureus* (including methicillin-resistant strains), coagulase-negative *Staphylococci*, as well as various *Streptococcus* and *Enterococcus* species such as faecalis and faecium. The Gram-negative bacteria *Escherichia coli*, *Pseudomonas aeruginosa*, *Acinetobacter baumanii*, *Klebsiella pneumoniae* and *oxytoca*, *Proteus mirabilis*, Enterobacter species, and *Burkholderia cepacian* are the most common. Hospital-acquired pneumonia typically arises from the aspiration or inhalation of contaminated aerosols. It may involve pathogens such as *Staphylococcus aureus*, *Pseudomonas aeruginosa*, *Streptococcus* and *Enterobacter* species, *Klebsiella oxytoca* and *pneumoniae*, as well as fungi *Candida* species, and multi-drug-resistant organisms.

According to the National Institutes of Health, the formation of pathogenic biofilms is linked to 65% of microbial infections and 80% of chronic infections [9]. The process of formation of pathogenic biofilms starts with bacterial bonding and invading on living or non-living surface, followed by colony development, where bacteria communicate and build resistance to the human immune system and antibiotics. There are two methods for preventing biofilm formation: preventing microbe growth with antimicrobial coatings and water purification and preventing microbe surface attachment through chemical or mechanical processes [10]. Polymicrobial biofilms present a significant challenge for treatment, as increasing antibiotic dosages often fails to combat them, particularly when they impair immune function and lead to persistent, severe infections on abiotic surfaces [11,12,13].

Catheter-associated urinary tract infections and central line-associated bloodstream infections frequently result in biofilm formation, facilitating growth and proliferation on external devices [14]. The most common species associated with these diseases are *S. aureus*, *E. coli*, *K. pneumoniae/oxytoca*, Enterococcus species, and *P. aeruginosa* [15]. *Escherichia coli* and *Pseudomonas aeruginosa* are the most common Gram-negative pathogens associated with biofilm formation on medical devices, posing significant challenges for antibiofilm strategies [16].

In our previous research, we synthesized new composite hydrogels made of bacterial cellulose and graphene quantum dots (BC-GQDs), which hold promise as wound dressings. These materials act not only as disinfectants but also as agents that promote the wound healing process [17]. Wound healing is a complex process driven by the intricate structure of the skin and the various biological processes occurring at different levels of its structure. This natural healing process is supported by good hygiene, proper nutrition, and adequate rest. However, bacterial infections are among the factors that can impede healing. Understanding the complexity of these factors is essential for developing effective agents, such as dressings, that can accelerate the healing process. Bacterial cellulose (BC) is a hydrogel material known for its crystalline nanofibrillar structure, excellent biocompatibility, large surface area, mechanical properties, and high water-holding ability [18,19,20]. As such, in addition to its antibacterial property, BC is very well used in our research as a carrier of many antibacterial compounds [21,22,23]. Graphene quantum dots (GQDs) are zero-dimensional disc-like nanoparticles derived from graphene or carbon nanotubes using a top-down method [24]. Their average diameter is 2–20 nm. Fitzgerald [25] reported their excellent chemical stability and high photoluminescence. Under blue light, GQDs exhibit antibacterial and phototoxic activity, with minimal cytotoxicity in the dark. Their antibacterial properties are attributed to their ability to produce singlet oxygen when exposed to visible light. As a photosensitizer (PS), GQDs show promise in antimicrobial photodynamic therapy (aPDT), which targets bacterial cells without harming healthy tissue. This non-invasive and painless treatment does not lead to bacterial resistance, reduces inflammation and bleeding, and specifically targets pathogenic bacterial cells without affecting healthy tissue [26]. The main effectiveness of the antimicrobial photodynamic therapy technique is eradicating multi-drug-resistant microorganisms [27,28,29,30]. The mechanism of this kind of therapy involves absorbing the visible light of PSs and transferring them from the ground state to an excited singlet state. The conversion to a long-lived excited triplet state is the next stage. All these states in the presence of oxygen generate reactive oxygen species, particularly singlet oxygen. The singlet oxygen oxidizes cellular components. Singlet oxygen’s ability to diffuse across cellular membranes makes it particularly effective against extracellular components such as those found in biofilms [31]. Nakono et al. (1998) and Tatsuzawa et al. (1998) reported its high oxidative damage efficiency against prokaryotic cells, while being completely non-toxic to eukaryotic cells [32,33].

In this study, we explore the potential of BC-GQD composite hydrogels in photodynamic therapy against pathogenic bacterial biofilms under blue and green light, based on the results of the previous investigation of the GQD’s potential in singlet oxygen production under different wavelength visible lights [34]. This research is the first report with in vitro results regarding the potential of visible light-induced BC-GQD composites to enhance antibiofilm efficacy. In addition to GQD’s possible ability to attach to bacterial membranes and cause destabilization [35], reactive oxygen species generation after exposure to these wavelengths is expected to have a dual effect on nosocomial-resistant strains. Additionally, this study investigates the effect of blue and green light-irradiated BC-GQD on cell migration at the molecular and cellular levels, demonstrating their potential role in promoting wound healing. These results could significantly enhance the effectiveness of photodynamic therapy approaches for combating biofilm-associated infections.

## 2. Results and Discussion

### 2.1. BC-GQD_Blue and BC-GQD_Green Surface Morphology and Chemical Composition

To visualize the surface of BC-GQD_control (ambient light illumination), BC-GQD_blue, and BC-GQD_green composite hydrogels, we used atomic force microscopy operated in tapping mode in air at ambient temperature, as shown in Figure 1. Figure 1a presents the optical photographs of BC hydrogels (left) and BC-GQD composite hydrogels (right). BC hydrogels are semi-transparent and white coloured, whereas BC composite hydrogels (BC-GQDs) are yellow colour appearance. Figure 1b shows top-view AFM images of BC-GQD_control (ambient light illumination), where as Figure 1c presents BC-GQD_blue (blue light illumination at 470 nm), and Figure 1d BC-GQD_green (green light illumination at 537 nm). This figure indicates the fibrous structure of BC-GQD composite hydrogels, and GQD can be spotted as a bright domain on the BC filaments, with average diameters of 80 nm (Figure 1a). The average diameter of the encapsulated GQD was 10 nm [34]. After the illumination of composite hydrogels with blue and green light at 30 nm, we could not detect any significant changes in the BC-GQD’s composite structure (Figure 1c,d).

In addition to surface morphology, it is very important to determine if light illumination can cause any changes in the chemical composition of irradiated samples as well. Thus, we conducted FTIR measurements of all specimens to detect any changes in the FTIR spectra of the illuminated specimen. The peak at 1581 cm^−1^ confirmed the encapsulation of GQD in the BC polymer as previously published [34]. Figure 2 represents the FTIR spectra of BC-GQD_control, BC-GQD_blue, and BC-GQD_green samples, respectively.

In these spectra, we detected the following peaks corresponding to O-H vibrations (3349 cm^−1^), C-H stretching vibrations (2890 cm^−1^), bending vibrations of the water molecules (1620 cm^−1^), C-O stretching vibrations (1162 cm^−1^, 1108 cm^−1^ and 1058 and cm^−1^), as well as C-H bending vibrations (897 cm^−1^). From Figure 2, we noticed that there were no shifts (up and down) in the FTIR spectra of the irradiated specimen compared to the control one, confirming no chemical changes in composites after exposure to green and blue light [36,37,38]. The existence of a peak at 1620 cm^−1^ signifies the higher amount of adsorbed water in these composite hydrogels [38].

Figure 3a,b show the PL spectra of BC-GQD illuminated with blue (λ = 470 nm) and green (λ = 537 nm) light for 30 min. After blue light irradiation, the highest PL emission could be observed at 370 and 395 nm excitation wavelengths, as shown in Figure 3a.

The PL emissions were up-shifted at 460 and 470 nm, respectively. After green light illumination, the highest PL emission could be observed at 395 nm excitation wavelength, as shown in Figure 3b. The up-shift in the PL emission spectrum was 75 nm. From the results obtained, we can conclude that the illumination with blue and green light contributed to the up-shift in PL emission spectra of about 70 nm compared to the excitation wavelength. It is interesting to note that visible light illumination causes the response of BC-GQD samples at 530 nm and at an excitation wavelength of 495 nm. The blue PL of GQDs observed in these composites is a feature of the quantum size effect and zig-zag edges [39]. Red-shifted PL is predominantly due to surface defects present on the basal plane and edges of the sp^2^ domain inside the sp^3^ matrix as well as the increase in size of aromatic π-conjugated domains [40].

### 2.2. EPR Measurement of Singlet Oxygen Formation

EPR was conducted to check the capability of BC-GQD hydrogel composites to produce reactive oxygen species after the illumination of blue and green light. All composites were illuminated with low-power (3 W) lamps at wavelengths of 470 nm and 537 nm overnight. Figure 4 shows the EPR intensity signal of BC-GQD_blue and BC-GQD_green hydrogel composites. As can be seen from this figure, both samples produced singlet oxygen. The EPR signal peak of BC-GQD_blue composites was almost more than double the intensity compared to the control, whereas the EPR signal peak of BC-GQD_green sample was 30% higher compared to the control.

In our previous investigation, we established that GQDs do not produce superoxide anions or hydroxyl radicals [39]. Thus, the primary process responsible for singlet oxygen generation is energy transfer from GQDs to molecular oxygen [41].

### 2.3. The Effect of BC-GQD_Blue and BC-GQD_Green Composites on Pathogenic Biofilms Followed by Atomic Force Microscopy (AFM) and Confocal Laser-Scanning Microscope (CLSM) Imaging of Biofilms

Biofilm formation is a key mechanism in bacterial resistance, offering protection against host immune defences and antibiotics [42,43]. Typically, in most antimicrobial photodynamic therapy studies, methodologies entail the preincubation of bacteria with PSs to facilitate its preservation via the biofilm matrix or the bacterial cell walls before illumination. This preservation may occur through bacterial attachment to the biofilm EPS matrix, interactions with the cell wall, or intracellular uptake and retention. Research has indicated that these steps are crucial for the effectiveness of the method. Without the preincubation of bacteria with the PS, antimicrobial photodynamic therapy is ineffective [44]. Since the GQDs are embedded deep within the matrix of the BC network rather than being located on the surface, time for preincubation was not required. In this research, the time needed to position the samples in the microplate wells was sufficient for the composites to be in contact with the biofilms. We propose that the primary mechanism by which these composite hydrogels eradicate biofilms is through the production of high levels of singlet oxygen. Table 1 presents the results of pathogenic bacteria biofilm reduction when BC-GQD composite hydrogels were placed on and after blue and green light illumination. The most effective was BC-GQD_blue on *P. aeruginosa*, with a 62% reduction, and BC-GQD_green on *E. coli* biofilms, with a 65% reduction (Table 1).

There are two intriguing aspects associated with cell wall structure and the capacity of bacterial clusters to form biofilms. Remarkably, these features directly correlate with the membrane wall structure of Gram-positive bacteria. In Gram-positive bacteria, glycopolymers, such as teichoic acids, play a critical role in host–cell attachment and biofilm development [45]. Conversely, because of their thinner peptidoglycan layer and extra outer membrane, Gram-negative bacteria show more tolerance to environmental challenges and drugs, thereby shielding them from external stresses. The teichoic acids in Gram-positive bacteria provide resistance to cationic antimicrobial peptides and influence autolysis, cell division, existence at higher temperatures, epithelial cell adhesion, as well as biofilm formation [46]. In this study, the smallest effect was observed with BC-GQD_blue on *S. aureus*, showing a biofilm reduction of less than 20%. However, MRSA biofilms were reduced by more than 40%, with BC-GQD_blue achieving a nearly 50% reduction.

The effect on pathogenic biofilms of BC-GQD_blue/green composites (Table 1) can be confirmed from the data obtained from the AFM images. Figure 5a displays the control sample’s AFM image and the continuous layer of *P. aeruginosa*’s biofilm. In contrast, following the illumination of BC-GQD composites placed on them (Figure 5b,c), respectively); there is a noticeable reduction in biofilm mass. The values of the average surface roughness (RMS) and surface area (Table 2) confirm all results presented in Table 1.

Live/dead staining of *P. aeruginosa* biofilms, followed by imaging on CLSM (Figure 6), showed nonhomogeneous pathogenic biofilms and effect of treatment with BC-GQD_green and BC-GQD_blue composite hydrogels.

The photo-induced significant decrease in surface roughness and surface area was obtained. The production of singlet oxygen destroys the bacterial cell membrane and damages intracellular constituents, leading to cell death and biofilm reduction [47,48].

### 2.4. BC-GQD_Blue and BC-GQD_Green Biocompatibility (MTT and NRU Assay)

Regarding HGF cells, the control and the BC-GQD composite hydrogel exposed to green or blue light exhibited no cytotoxicity for cells seeded directly on composites, as shown in Figure 7. The mitochondrial activity did not differ from the control. Light alone had no specific effect on HGF’s mitochondrial activity.

Taoufik et al. [49] reported that certain doses of blue light illumination moderately inhibit human gingival fibroblast proliferation. Specifically, using halogen (186 J/cm^2^), an LED (162 J/cm^2^), and a plasma arc (240 J/cm^2^) for 240, 180, and 120 s, respectively, resulted in this effect. As for our study, a low dose of blue light irradiation (3 J/cm^2^) for 30 min did not cause an adverse reaction to human gingival cells, suggesting that a low dose of blue light for a prolonged period of time is biocompatible.

Nakashima et al. [50] found that 72% of articles stated beneficial blue light therapeutic effects and 75% of articles reported similar effects with green light. The authors suggested a growth in cell proliferation modulating signal pathways due to a low-level increase in ROS production. Compared to these results, in our study, no proliferation was observed after 30 min of exposure to blue and green light. Further research could be directed to different irradiation settings to investigate possible cell proliferation effects under blue and green light irradiation.

These low irradiation doses (up to 10 J/cm^2^) showed viable human dermal fibroblasts cultured for two days [51]. These authors also showed safety to human fibroblast integrity with doses below 110 J/cm^2^.

In this study, the illumination lasted for 30 min, which was sufficiently short to induce singlet oxygen production via the GQDs, thereby activating their antibiofilm activity without affecting the viability of HGF cells. For HGF cells, antioxidant defence is crucial in preventing oxidative damage. Carotenoids, known for their ability to scavenge free radicals, play a vital role in protecting cells and tissues from oxidative stress. The gingival junctional epithelium, which is integral to periodontal health, is influenced by various agents, including carotenoids. These compounds are well documented for their powerful antioxidant properties, and they are capable of directly scavenging singlet molecular oxygen and peroxyl radicals. Additionally, they can synergistically interact with other antioxidants to further protect cells and tissues from oxidative damage [52,53]. Thus, bacteria and HGF cells employ distinct mechanisms to counteract oxidative stress [26,54].

### 2.5. In Vitro Wound Healing Analysis with the Scratch Assay

For the migratory potential of HGF cells (Figure 8), BC-GQD_blue stimulated an effect on cell migration, *p* < 0.05, as well as with blue light alone, *p* < 0.05. For BC-GQD_green and green light alone, no significant effect was observed in the observed period.

The significantly enhanced epithelialization and decreased wound size with blue light irradiation were shown in previous studies [55], affecting keratin expression. Likewise, low blue light illumination (470 nm, 5 J/cm^2^) of human dermal fibroblasts promoted in vitro wound healing, modulating the synthesis of cytokines, growth factors, total protein, and the most important collagen [56]. In this study, *eNOS*, *MMP*, and *Vimentin* gene expressions in HGFs were evaluated following irradiation with blue and green light alone and in combination with BC-GQD composite hydrogels. The results are presented in the next section.

### 2.6. Gene Expression of eNOS, MMP9, and Vimentin

For HGF cells, the treatment with BC-GQD_blue significantly stimulated the expression of *eNOS* (*p* < 0.0001) and *MMP9* (*p* < 0.0001), while for *Vimentin*, a trend of stimulated expression was observed for BC-GQD_blue and blue light alone but with no significant difference to the control (Figure 9). On the other hand, BC-GQD_green had a limited effect on the expression of *eNOS* and *MMP9*, while significantly downregulated the expression of *Vimentin*, *p* < 0.01.

The gene expression of key proteins such as endothelial nitric oxide synthase (*eNOS*), matrix metallopeptidase 9 (*MMP9*), and *Vimentin*, besides the crucial role in supporting angiogenesis, influence the viability and migration of cells. *eNOS* produces low levels of nitric oxide, which plays a key role in mediating various physiological functions, such as upregulating *MMP9* to promote progenitor cell recruitment and vessel formation. Nitric oxide serves as a central mediator, driving endothelial cell migration, proliferation, and the expression of angiogenic factors that are essential for tissue repair and regeneration.

The obtained results of these gene expressions supported the scratch assay results, where BC-GQD_blue promoted wound healing. The decreased gene expression of *Vimentin* with BC-GQD_green might be a good starting point for anti-cancer effect investigation. The activation of genes could be dose-dependent, as contrary results have been reported when different doses of blue or green light were applied in melanocytes [57]. Green light is known to affect the ERK signalling pathway [58] that is strongly linked to *Vimentin* expression, which is upregulated in numerous cancers [59].

## 3. Materials and Methods

### 3.1. The Synthesis of GQD, Followed by BC Loading

GQD suspension was prepared with the already used method, described in detail by [26]. BC hydrogel samples were synthesized via bacterial culture of *Komagataeibacter oboediens* IMBG180 (the synthesis detailed previously in [21]). A dipped method was used for BC-GQD composite hydrogel sample preparation in 0.2 and 2 mg/mL GQD suspensions [17].

The composites were subjected to green (537 nm, 3 W, 30 min; BC-GQD_green) and blue (470 nm, 3 W, 30 min; BC-GQD_blue) light. The distance between the sample surface and a lamp was 20 cm, and the irradiation light was homogeneous. Temperature changes near the samples and plates were not observed. Composite hydrogels under ambient light (BC-GQD_control) were the control sample.

### 3.2. Atomic Force Microscopy of BC-GQD Samples

The surface morphology of all samples was investigated via atomic force microscopy (AFM-Quesant, Santa Barbara, CA, USA), operating in tapping mode in the air at room temperature with standard silicon tips NanoAndMore GmbH (Wetzlar, Germany) with a force constant of 40 N/m. The Q-WM300 probe, a rotated monolithic silicon probe for non-contact high-frequency applications, was used. All samples were dried at 50 °C in a vacuum furnace (Memmert, Schwabach, Germany) before measurements. The Gwyddion software 2.64 was used for AFM image analysis [60].

### 3.3. Fourier Transform Infrared Spectroscopy (FTIR) Measurements

FTIR spectra of all investigated samples (BC-GQD_control, BC-GQD_green, and BC-GQD_blue) were obtained using a Nicolet iN10 Thermofisher Scientific infrared microscope. It operated in ATR mode, and spectra were recorded from 400 to 4000 cm^−1^ at ambient conditions. During measurements, the spectral resolution was 4 cm^−1^. All samples were dried at 50 °C in a vacuum furnace (Memmert, Schwabach, Germany) before measurements.

### 3.4. Photoluminescence (PL) Measurements

To measure the PL of all samples, a Fluorog spectrofluorometer (Horiba, Kyoto, Japan) operating at ambient conditions was used. All samples were dried at 50 °C in a vacuum furnace (Memmert, Schwabach, Germany) before measurements. Each sample before PL measurement was put in a holder for solid samples. The excitation wavelengths were in the range of 320–495 nm.

### 3.5. Singlet Oxygen Detection with Electron Paramagnetic Resonance (EPR) Analysis

The EPR measurements were conducted on a Spectrometer MiniScope 300, Magnettech, Berlin, Germany, to check for singlet oxygen production of BC-GQD_green and BC-GQD_blue composite hydrogels. The microwave power was 1 mW (microwave attenuation of 20 dB), with a modulation amplitude of 0.2 mT. The instrument was operating at a nominal frequency of 9.5 GHz. As a spin trap, we used 2,2,6,6-tetramethylpiperidine (TEMP; ≥99%; Merck, Darmstdt, Germany). With ^1^O_2_, TEMP molecules quickly react and form a stable, EPR-active product, TEMP-^1^O_2_ (TEMPO). The samples were immersed in 30 mM TEMP ethanol solution and illuminated with low-power (3 W) lamps at wavelengths 470 nm and 537 nm overnight.

### 3.6. Pathogenic Biofilm Reduction with Green and Blue Light Irradiation—In Vitro Test

The antibiofilm activity of photo-induced BC-GQD hydrogels was conducted according to a previously established procedure [61]. In brief, pathogenic bacteria-referent strains, namely *Staphylococcus aureus* (ATCC 25923), methicillin-resistant *Staphylococcus aureus* (MRSA, ATCC 43300), *Escherichia coli* (ATCC 25922), and *Pseudomonas aeruginosa* (ATCC 27853), were set to form biofilm. The 5 µL bacterial suspension, set according to ISO 20776-1:2016 [62] containing 5 × 10^5^ CFU/mL was set in a liquid medium with 100 µL TSB (Becton, Dickinson and Company, Franklin Lakes, NJ, USA GTIN: 00382902924388) supplemented with 2% dextrose, incubated overnight aerobically at 37 °C, followed by rinsing each well with 100 μL three times with sterile saline to remove planktonic bacteria.

Prepared biofilm plates were added to the experiment by adding 100 μL of fresh medium and disc composites in the wells. To activate the photoactive effect, the plates were illuminated with green (537 nm) or blue light (470 nm) for 30 min, with the 3 W lamp positioned 20 cm away, without temperature changes near the samples and plates, as shown in Appendix A. After illumination, the plates were placed overnight at 37 °C.

Finally, to assess the antibiofilm effect, the wells were rinsed again with 100 μL of sterile saline three times and incubated for 2 h with 3% 2,3,5-Triphenyltetrazolium chloride (TTC) salt solution (Sigma, Saint Louis, MO, USA, art. T8877) in TSB with 2% dextrose. Furthermore, the dissolution of formazan crystals was conducted with 100 µL of DMSO (Zorka, Šabac, Serbia) followed by the determination of absorbance at 450 nm with a microplate reader (Tecan Sunrise, Tecan, Grödig, Austria). For the control, the BC discs without GQDs were used (blank). All tests were performed twice. Biofilm eradication (%) was calculated and compared to BC-GQD absorbance, with blanks using the following equation: biofilm eradication = (((OD(blank) − OD(treatment))/OD(blank)) × 100 [61]. The optical images of wells with *P. aeruginosa* biofilms before and after staining with TTC and crystal violet are presented in Appendix A. The crystal violet assay was performed by following a previously published method [63].

### 3.7. Imaging of Pathogenic Biofilms with Atomic Force Microscopy (AFM), Confocal Laser-Scanning Microscope (CLSM) and Fluorescent Microscope

AFM, as an influential method for imaging microbial surfaces, was utilized in this study. After the treatments with composites, the pathogenic biofilms of *Pseudomonas aeruginosa* were air dried on microscope glass slides and examined in the air with tapping mode via an AFM Quesant microscope (Ambios Technology, Santa Cruz, CA, USA). All images with 512 × 512 resolution were obtained at 2 Hz. The Gwyddion 2.64 software was used for calculations of the most relevant properties of biofilms: average surface roughness and surface area [60].

The bacterial biofilm of *Pseudomonas aeruginosa* was stained with fluorescein diacetate (8 µg/mL) and propidium iodide (1 µg/mL) in PBS for 15 min and then visualized on a CLSM (LSM 510, Carl Zeiss GmbH, Jena, Germany). Fluorophores were excited with 488 and 543 nm lasers; a water-immersion objective 40×/NA 0.8 was used, and the pinhole was set to 7 µm for both detection channels.

The bacterial biofilm of *Pseudomonas aeruginosa* was also stained with propidium iodide using the method described previously [64]. Images were captured on a Zeiss Axiovert 5 fluorescent microscope using the Zeiss Zen 3.8 software, with 20× and 40× magnification (Carl Zeiss AG, Oberkochen, Germany) (Appendix A).

### 3.8. In Vitro Investigation on HGF

All experiments were designed to investigate cells’ response to different light exposures using composite hydrogels, providing insights into mitochondrial activity, wound healing capabilities, and gene expression profiles. Human gingival fibroblasts (HGFs) were used in the study, isolated from human gingival tissues described by Mančić et al. The study was conducted in accordance with the Declaration of Helsinki, and the pro-tocol was approved by the Ethics Committee of the School of Dental Medicine, Univer-sity of Belgrade, Serbia (Protocol number 36/2) [65]. All trials were conducted as defined in Zmejkoski et al. [17]. All chemicals used for the biocompatibility study are of high purity, sterile, and used under sterile conditions under a laminar hood.

In brief, to assess mitochondrial activity, hydrogel discs (6 mm diameter) were placed in 96-well plates with a DMEM/F12 medium added with 10% fetal bovine serum (FBS) and 2% ABAM as an antibiotic/antimycotic reagent (Thermo Fisher Scientific, Waltham, MA, USA). Cells (5 × 10^3^/well) were seeded the next day and incubated in the medium at 37 °C with 5% CO_2_. The plates were illuminated with green (537 nm, 3 W, 30 min) or blue (470 nm, 3 W, 30 min) light at a distance of 20 cm for 30 min; non-exposed cells served as a control. Also, there was a control group of cells without any treatment. The discs without cells served as a control for the unspecific binding of used dyes in tests. After 24 h, mitochondrial activity was evaluated with the MTT assay (0.5 mg/mL, Sigma) as an indirect test of cell viability. As for a direct test of cell membrane integrity, the neutral red uptake (NRU) assay (NRU solution, 40 μg/mL; Sigma Aldrich, Burlington, MA, USA) was used as previously described by Repetto et al. [66].

For wound healing analysis, through the promotion of migration, cells (5 × 10^4^ cells/well) were seeded and left to reach a confluence of 80%. For further process, cells were cultivated in a DMEM/F12 medium supplemented with 2% FBS and 1% ABAM. Scratches were made, and fresh medium was added. Hydrogel disc samples were positioned in the wells, and corresponding light exposure conditions were administered. Images were captured at 0 and 24 h using an inverted microscope (BIB-100/T, BOECO, Hamburg, Germany) with an HDCE-90D camera (BOECO, Germany), 100× magnification. Scratched areas were analyzed with Fiji/ImageJ 1.54g, and the results were calculated as the difference between the area of scratch at 0 h and 24 h for each well, respectively. The experiment was performed in triplicate.

For angiogenesis promotion, the relative gene expression levels of *eNOS*, *MMP9*, and *Vimentin* were examined using hydrogel discs (9.5 mm diameter, 1 mm height) in 6-well plates. Cells (1 × 10^5^ cells/well) were incubated for 24 h, and corresponding light exposure conditions were applied. After 72 h, the RNA was isolated from HGF using the TRIzol reagent according to the manufacturer’s instructions (Thermo Fisher Scientific). The concentration of RNA was determined using a microvolume spectrophotometer (BioSpec-nano Microvolume UV–Vis Spectrophotometer; Shimadzu Scientific Instruments, Columbia, MD, USA). Complementary DNA was synthesized from 1 μg of total RNA using the Thermo Scientific RevertAid First Strand cDNA Synthesis Kit (Thermo Fisher Scientific) in the presence of oligo(dT) primers. The amplification of the selected targeted genes’ regions was assessed with the SensiFASTtm SYBR^®^ Hi-ROX Kit (Bioline, London, UK). For normalization, glyceraldehyde 3-phosphate dehydrogenase (GAPDH) was used. The specific human primer sequences used for mRNA expression were as follows (5′-3′): *eNOS*: forward CACCGCTACAACATCCTG, reverse GCCTTCTGCTCATTCTCC; *MMP9*: forward GAACAAATACAGCTGGTTCC, reverse TACCCTATGTACCGCTTCAC; *Vimentin*: forward TCTACGAGGAGGAGATGCGG, reverse GGTCAAGACGTGCCAGAGAC; GAPDH: forward TCATGACCACAGTCCATGCCATCA, reverse CCTGTTGCTGTAGCCAAATTCGT. The final volume of the PCR reaction was 15 μL, containing SYBR Green qPCR Master Mix, 200 nmol/L of each primer, and 2 μL of cDNA. All samples were running in duplicate in three independent experiments for each sample. Real-time PCR (qPCR) was performed using the Line gene K fluorescence quantitative PCR detection system (BIOER Technology Co., Hangzhou, China). After a qPCR run, a melting curve analysis was performed to confirm the specificity of the run. The results obtained from each run were threshold cycle (Ct) values, the cycle number at which the fluorescence level goes over a fixed threshold. Relative gene expression values were calculated using the 2^−ΔCt^ method [67]. The ratio between *eNOS*, *MMP9*, and *Vimentin* expression and GAPDH expression was used as the relative expression level of mRNA. Furthermore, relative gene expression is presented as a relative fold change in comparison to the control group. The specificity of qPCR assays is presented in Appendix A.

## 4. Conclusions

In this study, a novel photoactive BC-GQD composite hydrogels was fabricated using the facile and green method, demonstrating promise in combating nosocomial infections. This study explored the potential of photodynamic therapy by subjecting the composites to short-term irradiation with blue (470 nm) and green (537 nm) light. FTIR, PL, and EPR analyses confirmed no chemical changes in composites after illumination with blue and green light. FTIR, PL, and EPR analyses confirmed no chemical changes in composites after illumination with blue and green light, a 70 nm up-shift in the photoluminescence emission spectra compared to the excitation wavelength, and singlet oxygen production from 30% to two times higher than control samples.

The composite under blue and green light exhibited a significant antibiofilm effect against *E. coli*, MRSA, and *S. aureus*, with the most effectiveness on *P. aeruginosa*, which was confirmed with a microtiter biofilm reduction assay and AFM images and supported by changes in biofilms surface area and average surface roughness. Furthermore, the photoactive BC-GQD composite exhibited no cytotoxic effect with the promotion of wound healing through the acceleration of cells’ migration and increased *eNOS* and *MMP9* gene expression of HGF cells. Since our previous study already demonstrated good wound fluid absorption and water retention of BC-GQD composites under ambient light, which potentially provide good moisture and additional faster epithelization, further research will focus on conducting experiments on animals to investigate in vivo performance of the composite in photodynamic therapy. Based on the results obtained, their strong effectiveness against *Pseudomonas aeruginosa*, one of the most common and transmissible hospital pathogens, greatly enhances their potential. The dual effect as antibiofilm agents and wound healing facilitators suggests the high applicability of BC-GQD as a photodynamic therapy agent in a novel approach to preventing and treating nosocomial infections, positioning them as super potent agents for wound remediation.

## Figures and Tables

**Figure 1 ijms-26-01053-f001:**
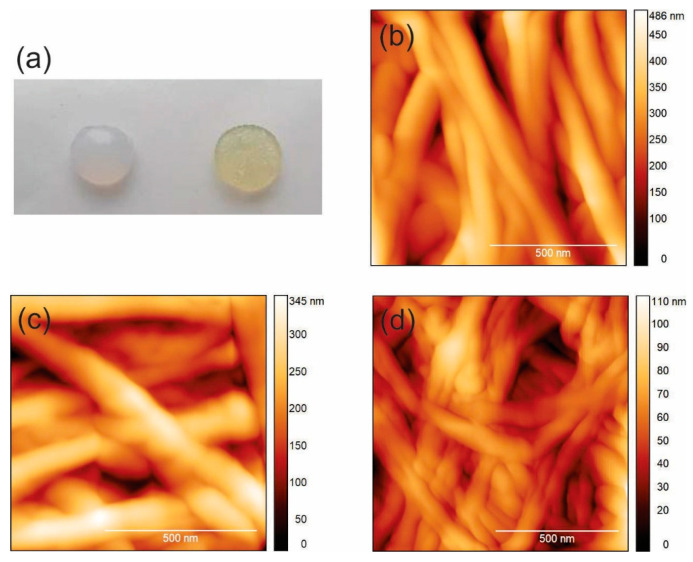
(**a**) Optical photographs of BC (**left**) and BC-GQD (**right**) samples; top-view AFM images of (**b**) BC-GQD_control (ambient light illumination), (**c**) BC-GQD_blue (blue light illumination at 470 nm), and (**d**) BC-GQD_green (green light illumination at 537 nm).

**Figure 2 ijms-26-01053-f002:**
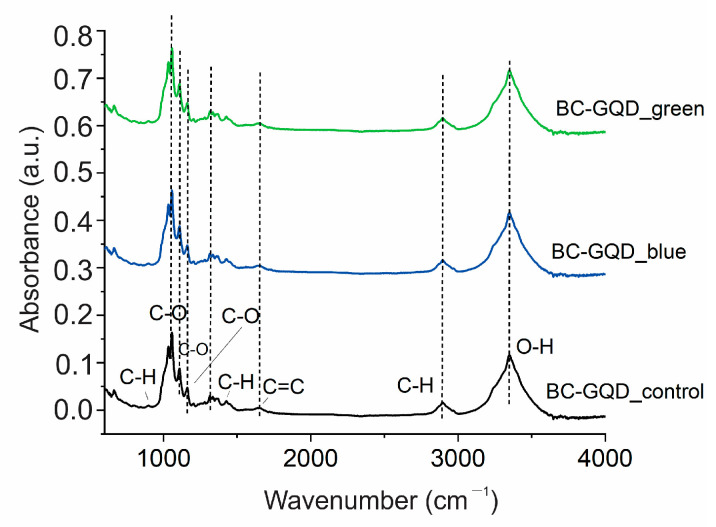
FTIR spectra of BC-GQD_control (ambient light illumination; black curve), BC-GQD_blue (blue light illumination at 470 nm; blue curve), and BC-GQD_green (green light illumination at 537 nm; green curve).

**Figure 3 ijms-26-01053-f003:**
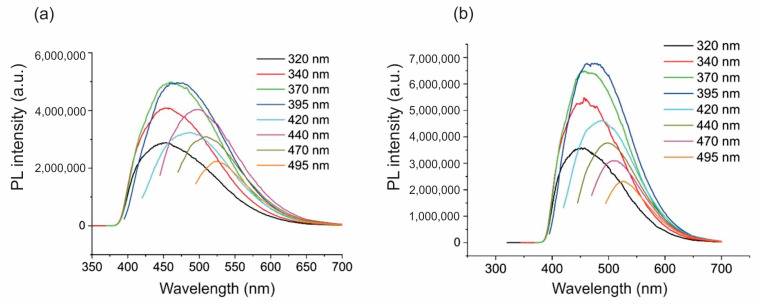
(**a**) PL intensity of BC-GQD hydrogel composites irradiated with blue light at 470 nm; (**b**) PL intensity of BC-GQD hydrogel composites irradiated with green light at 537 nm.

**Figure 4 ijms-26-01053-f004:**
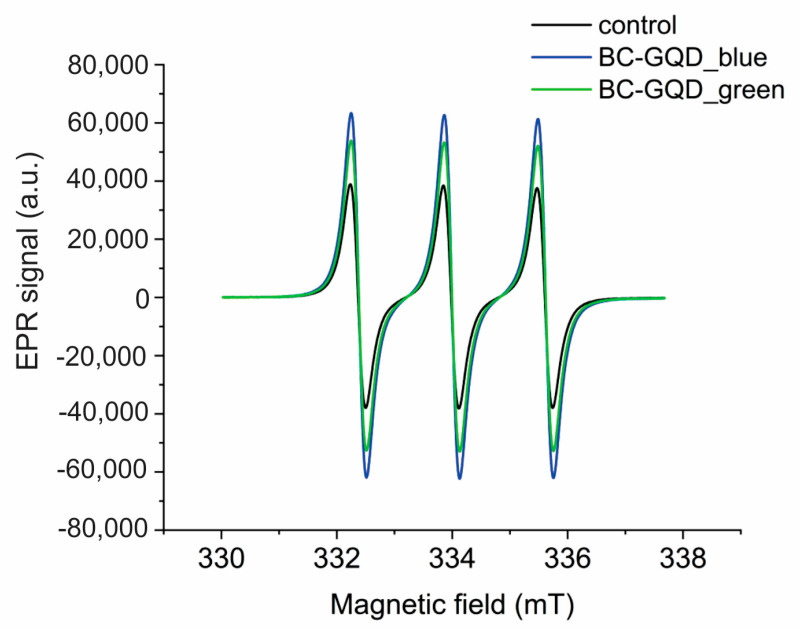
EPR spectra of BC-GQD composite hydrogel samples illuminated with blue and green light (at 470 nm and 537 nm, respectively).

**Figure 5 ijms-26-01053-f005:**

Top-view AFM images of (**a**) well-formed *Pseudomonas aeruginosa* biofilm, (**b**) treated with BC-GQD_blue hydrogel composites and (**c**) treated with BC-QGD_green. The scanned surface for each image is 15 × 15 µm^2^.

**Figure 6 ijms-26-01053-f006:**
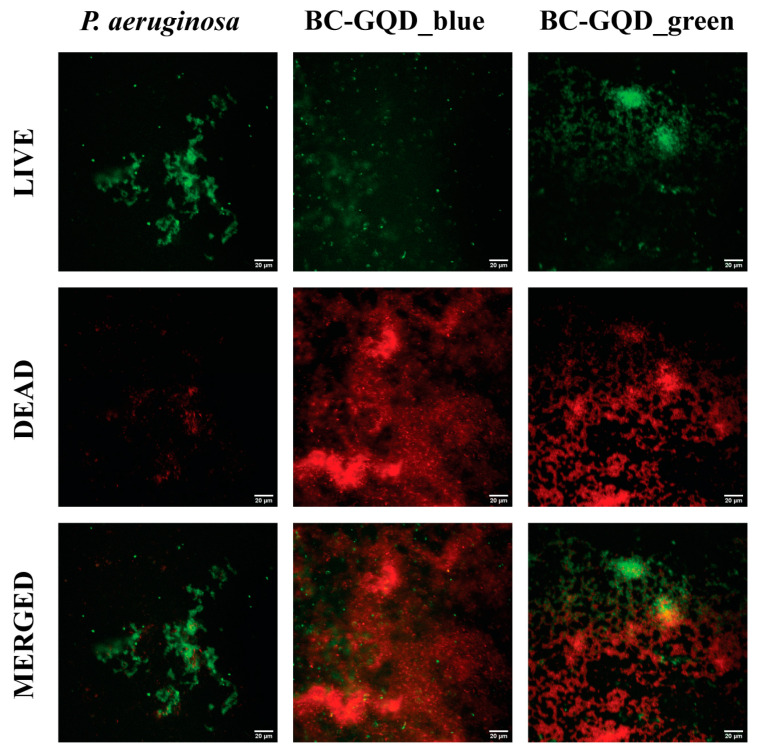
CLSM images of *Pseudomonas aeruginosa*’s biofilm before and after the treatment with the BC-GQD_green and BC-GQD_blue hydrogel composites. The scale bars are 20 µm.

**Figure 7 ijms-26-01053-f007:**
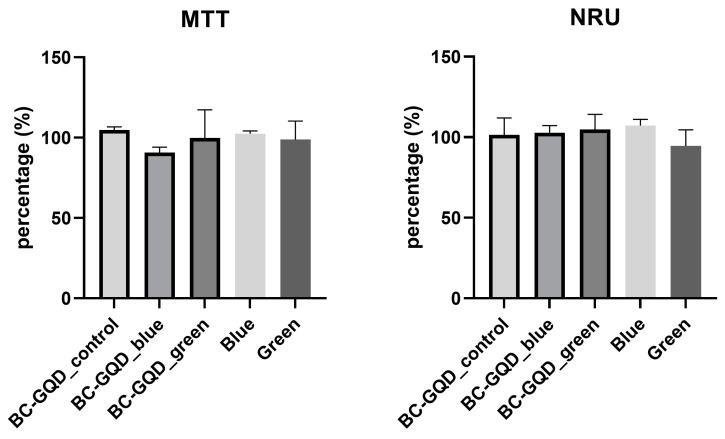
Biocompatibility tests—MTT and NRU. Mitochondrial activity and the proliferation of the cells seeded directly on the composites under ambient, blue, and green light illumination (BC-GQD_control, BC-GQD_blue, and BC-GQD_green, respectively) as well as with blue or green light alone. The values of the error bars are standard deviations; Kruskal–Wallis test was used, *p* < 0.05.

**Figure 8 ijms-26-01053-f008:**
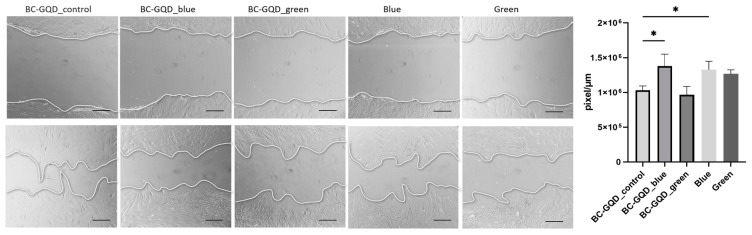
HGF migration induced via the application of BC-GQD_control, BC-GQD_blue, and BC-GQD_green at the following time intervals: start point and after 24 h (top and bottom row, respectively). Scale bar is 700 μm; Kruskal–Wallis test was used, * *p* < 0.05.

**Figure 9 ijms-26-01053-f009:**
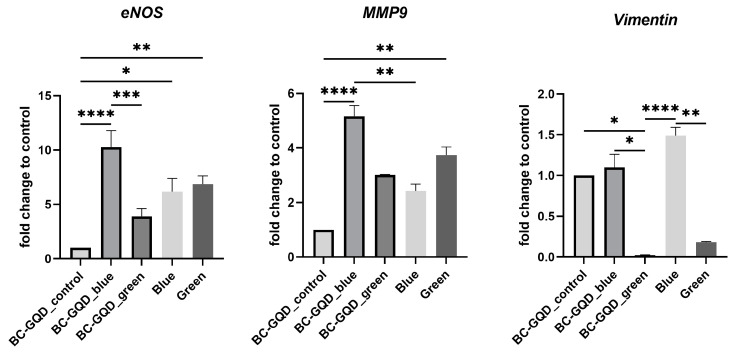
*eNOS*, *MMP9*, and *Vimentin* relative gene expression after 72 h of HGF in the presence of the BC-GQD_control, BC-GQD_blue and BC-GQD_green composites, as well as blue and green light alone; Kruskal–Wallis test was used (* *p* < 0.05, ** *p* < 0.01, *** *p* < 0.001, and **** *p* < 0.0001). The values of the error bars are standard deviations.

**Table 1 ijms-26-01053-t001:** Microtiter pathogenic biofilm reduction assay. The values are presented in %.

Bacteria	BC-GQD_Blue	BC-GQD_Green
*S. aureus*	19%	51%
MRSA	48%	42%
*E. coli*	57%	65%
*P. aeruginosa*	62%	45%

**Table 2 ijms-26-01053-t002:** Gwyddion 2.64 software calculation of *P. aeruginosa* biofilms’ RMS and surface area. Values are presented in nm and µm^2^, respectively.

Bacteria	Blanc	BC-GQD_Blue	BC-GQD_Green
	RMS [nm]		
*P. aeruginosa*	119.52 ± 24.86	51.26 ± 13.60	53.14 ± 22.89
	Surface area [µm^2^]		
*P. aeruginosa*	166.55 ± 16.93	110.65 ± 2.93	121.69 ± 9.65

## Data Availability

Data are contained within the article and Appendix A.

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
