# Peer review of "Graphene Quantum Dots in Bacterial Cellulose Hydrogels for Visible Light-Activated Antibiofilm and Angiogenesis in Infection Management"

_ijms, 2025, doi:10.3390/ijms26031053_

Round 1

Reviewer 1 Report

Comments and Suggestions for Authors

The manuscript titled “Graphene quantum dots in bacterial cellulose hydrogels for visible light-activated antibiofilm and angiogenesis in infection management” by Zmejkoski, D.Z.; et al. is a scientific work where the authors designed hydrogels made of bacterial cellulose and graphene quantum dots with antimicrobial capability after light exposure by the formation of reactive oxygen species. The hydrogels were fully characterized. For it, many complementary techniques were devoted in this research. The most relevant outcomes found in this research could open new gates in the development of smart materials agains pathogen infections. The manuscript is generally well-written and this is a topic of growing interest.

However, it exists some points that need to be addressed (please, see them below detailed point-by-point) to improve the scientific quality of the submitted manuscript paper before this article will be consider for its publication in the International Journal of Molecular Sciences.

1) The authors should consider to add the term “human gingival fibroblasts” in the keyword list.

2) “Nosocomial infections develop within a healthcare facility (…) These infections include urinary tract infections (…) respiratory pneumonia (…) skin and gastrointestinal infections, and surgical site wound infections” (lines 36-40). Could the authors provide quantitative data insights according to the worldwide global burdens of incidence concerning the above described nosocomial infections? This will significantly aid the potential readers to better understand the significance of this devoted research.

3) “In our previous research, we have synthesized new composite hydrogels made of bacterial cellulose and graphene quantum dots (…) Bacterial cellulose (BC) is a hydrogel material (…) excellent biocompatibility, offering large surface area, and high water-holding ability” (lines 85-95). First the abbreviation related to the term bacterial cellulose should appear the first time that the full-name appears in the main manuscript body text. Then, it should be remarkable to also mention the excellent mechanical properties exhibited by cellulose materials [1] that make them suitable for many Industrial applications [2].

[1] https://doi.org/10.1016/j.ijbiomac.2019.10.074

[2] https://doi.org/10.3390/ijms25010069

4) “2.1. BC-GQD_blue and BC-GQD_green surface morphology and chemical composition” (lines 130-174). Did the authors characterize diameter of the graphene quantum dots used in this research (e.g. by dynamic light scattering measurements)?

5) Figure 4 (line 229). Could the authors provide the 2D-images without any tilt? This will benefit to have a more complete outlook of the tested sample topology.

6) Table 2 (line 242). Could the dehydration effect due to the measurements taken in air conditions have an impact on the roughness parameters displayed in this Table. A brief discussion should be furnished in this regard.

7) Figure 6 (line 249). Similarly to the Fig. 7 and Fig. 8 some statistical analysis needs to be conducted in order to discern if the observed differences are statistically significant among the examined conditions.

8) “4. Conclusions” (lines 456-480). This section perfectly remarks the most relevant outcomes found by the authors in this work and also the promising future prospectives in this field. It may be advisable to add a brief statement to remark the potential future action lines to pursue the topic covered in this topic.

Author Response

Dear Sir,

Thank you for careful reading of our manuscript. We accepted all your suggestions and corrected manuscript as requested. All changes in the manuscript have been highlighted. Our answers are given below:

Referee 1

The manuscript titled “Graphene quantum dots in bacterial cellulose hydrogels for visible light-activated antibiofilm and angiogenesis in infection management” by Zmejkoski, D.Z.; et al. is a scientific work where the authors designed hydrogels made of bacterial cellulose and graphene quantum dots with antimicrobial capability after light exposure by the formation of reactive oxygen species. The hydrogels were fully characterized. For it, many complementary techniques were devoted in this research. The most relevant outcomes found in this research could open new gates in the development of smart materials agains pathogen infections. The manuscript is generally well-written and this is a topic of growing interest.

However, it exists some points that need to be addressed (please, see them below detailed point-by-point) to improve the scientific quality of the submitted manuscript paper before this article will be consider for its publication in the International Journal of Molecular Sciences.

1) The authors should consider to add the term “human gingival fibroblasts” in the keyword list.

 We inserted the term “human gingival fibroblasts” in the keyword list.

2) “Nosocomial infections develop within a healthcare facility (…) These infections include urinary tract infections (…) respiratory pneumonia (…) skin and gastrointestinal infections, and surgical site wound infections” (lines 36-40). Could the authors provide quantitative data insights according to the worldwide global burdens of incidence concerning the above described nosocomial infections? This will significantly aid the potential readers to better understand the significance of this devoted research.

 We inserted requested changes in the manuscript, pages 1-2.

3) “In our previous research, we have synthesized new composite hydrogels made of bacterial cellulose and graphene quantum dots (…) Bacterial cellulose (BC) is a hydrogel material (…) excellent biocompatibility, offering large surface area, and high water-holding ability” (lines 85-95). First the abbreviation related to the term bacterial cellulose should appear the first time that the full-name appears in the main manuscript body text. Then, it should be remarkable to also mention the excellent mechanical properties exhibited by cellulose materials [1] that make them suitable for many Industrial applications [2].

[1] https://doi.org/10.1016/j.ijbiomac.2019.10.074

[2] https://doi.org/10.3390/ijms25010069

We inserted the requested references in the manuscript (Introduction section, references 19,20) and BC abbreviation is mentioned the first time appeared in the text.

 4) “2.1. BC-GQD_blue and BC-GQD_green surface morphology and chemical composition” (lines 130-174). Did the authors characterize diameter of the graphene quantum dots used in this research (e.g. by dynamic light scattering measurements)?

We characterized the diameter of GQDs in our previous research by determination of particle size distribution by AFM (Markovic, Z. et al., J Appl Polym Sci 2021, e51996). We determined that average diameter of GQDs encapsulated in BC hydrogel is 10 nm.

 5) Figure 4 (line 229). Could the authors provide the 2D-images without any tilt? This will benefit to have a more complete outlook of the tested sample topology.

We inserted new 2D top view AFM images (new Figure 5).

 6) Table 2 (line 242). Could the dehydration effect due to the measurements taken in air conditions have an impact on the roughness parameters displayed in this Table. A brief discussion should be furnished in this regard.

At the moment, we cannot afford the determination of the roughness of bacteria biofilm by using wet cell. The only possibility to observe bacterial biofilm morphology is to dry them on microscopic glass and recording their surface morphology by AFM. According to literature data, filamentous EPS (extracellular polymeric substances, forming the matrix of microbial biofilm structures) observed in bacterial samples prepared and imaged by traditional electron microscopy methods have been suggested to be involved, directly or indirectly, in physical interactions and aggregation, cell-to-surface attachment by tethering via thin adhesion threads, extracellular electron transfer, and possibly de facto electrical signaling within the structurally integrated communities (Dohnalkova, A.C. et al, Applied and Environmental Microbiology , Feb. 2011, p. 1254–1262, doi:10.1128/AEM.02001-10). As for roughness of bacteria biofilm in the hydrated state, we could not find any literature data related to that. There are data only related to morphology, structure and cell elasticity (Dufrene, Y., J Bacteriol. 2002 Oct;184(19):5205–5213. doi: 10.1128/JB.184.19.5205-5213.2002). In this study, we want to check if visible light (blue and green separately) affects the reduction of bacteria biofilms. We got good results by applying biological method and we wanted to document results obtained by visualization of bacteria biofilm (AFM). Thus, we measured surface roughness and surface area and pointed out that these data changed after illumination by visible light. Later, we used CLSM method as well as fluorescence microscopy of biofilm to prove our previous results.

 7) Figure 6 (line 249). Similarly to the Fig. 7 and Fig. 8 some statistical analysis needs to be conducted in order to discern if the observed differences are statistically significant among the examined conditions.

The applied statistical test was added in Figure 6 (new Figure 7) caption as requested. No significant difference among tested groups was found.

 8) “4. Conclusions” (lines 456-480). This section perfectly remarks the most relevant outcomes found by the authors in this work and also the promising future prospective in this field. It may be advisable to add a brief statement to remark the potential future action lines to pursue the topic covered in this topic.

Our further research will focus on conducting experiments on animals to investigate in vivo performance of the composite in photodynamic therapy as stated in the conclusion.

Best regards,

Biljana Todorovic Markovic

Reviewer 2 Report

Comments and Suggestions for Authors

Reviewer report on manuscript ijms-3420466

Danica Z. Zmejkoski et al.Graphene quantum dots in bacterial cellulose hydrogels for visible light-activated antibiofilm and angiogenesis in infection management

In present study, A novel bacterial cellulose-based composite hydrogel with graphene quantum dots (BC-GQD) was developed for photodynamic therapy using blue and green light (BC-GQD_blue and BC-GQD_green) to target pathogenic bacterial biofilms. This approach aimed to address complications in treating nosocomial infections and combating multi-drug resistant organisms. Short-term illumination (30 minutes) of both BC-GQD samples led to singlet oxygen production and a reduction in pathogenic biofilms. Significant antibiofilm activity (>50% reduction) was achieved against Staphylococcus aureus and Escherichia coli with BC-GQD_green, and against Pseudomonas aeruginosa with BC-GQD_blue. Atomic force microscopy images revealed a substantial decrease in biofilm mass, accompanied by changes in surface roughness and area, further confirming the antibiofilm efficacy of BC-GQD under blue and green light, without any observed chemical alterations.

The manuscript can be accepted after major revision.

Overall, the quality of the work is good, however I point out several questions to help the authors improve the manuscript before publication.

Questions/comments:

1.      I recommend restructuring the manuscript and move Figure S1 to the main text.

2.      More details to the section “3. Materials and Methods” should be added, including information about AFM, FTIR, PL, EPR analysis.

3.       The FTIR spectra are not very good justified. There are not up-to-date references (2024) for choice of the components. The references to up-to date 2024 investigations in this field should be added, e.g. [Carbon 2022, 196, 264], [Nanomaterials 2023, 13, 23], [Nanomaterials 2023, 13, 1730], and references there.

Author Response

Dear Sir,

Thank you for careful reading of our manuscript. We accepted all your suggestions and corrected manuscript. All changes in the manuscript have been highlighted. Our answers are given below:

Referee 2

Danica Z. Zmejkoski et al. “Graphene quantum dots in bacterial cellulose hydrogels for visible light-activated antibiofilm and angiogenesis in infection management”

In present study, A novel bacterial cellulose-based composite hydrogel with graphene quantum dots (BC-GQD) was developed for photodynamic therapy using blue and green light (BC-GQD_blue and BC-GQD_green) to target pathogenic bacterial biofilms. This approach aimed to address complications in treating nosocomial infections and combating multi-drug resistant organisms. Short-term illumination (30 minutes) of both BC-GQD samples led to singlet oxygen production and a reduction in pathogenic biofilms. Significant antibiofilm activity (>50% reduction) was achieved against Staphylococcus aureus and Escherichia coli with BC-GQD_green, and against Pseudomonas aeruginosa with BC-GQD_blue. Atomic force microscopy images revealed a substantial decrease in biofilm mass, accompanied by changes in surface roughness and area, further confirming the antibiofilm efficacy of BC-GQD under blue and green light, without any observed chemical alterations.

 The manuscript can be accepted after major revision.

Overall, the quality of the work is good, however I point out several questions to help the authors improve the manuscript before publication.

Questions/comments:

1.I recommend restructuring the manuscript and move Figure S1 to the main text.

We moved Figure S1 from supporting information in the main text and labelled figure as Figure 1.

2.More details to the section “3. Materials and Methods” should be added, including information about AFM, FTIR, PL, EPR analysis.

We added more text about AFM, FTIR, PL, and EPR analysis.

3.The FTIR spectra are not very good justified. There are not up-to-date references (2024) for choice of the components. The references to up-to date 2024 investigations in this field should be added, e.g. [Carbon 2022, 196, 264], [Nanomaterials 2023, 13, 23], [Nanomaterials 2023, 13, 1730], and references there

We added the requested references in the manuscript (references 37,38).

Best regards

Biljana Todorovic Markovic

Round 2

Reviewer 2 Report

Comments and Suggestions for Authors

Manuscript can be accepted in revised form.